# MULTI-STEP DECENTRALIZED DOMAIN ADAPTATION

## ABSTRACT

Despite the recent breakthroughs in unsupervised domain adaptation (uDA), no prior work has studied the challenges of applying these methods in practical machine learning scenarios. In this paper, we highlight two significant bottlenecks for uDA, namely excessive centralization and poor support for distributed domain datasets. Our proposed framework, MDDA, is powered by a novel collaborator selection algorithm and an effective distributed adversarial training method, and allows for uDA methods to work in a decentralized and privacy-preserving way.

## 1 Introduction

In practical machine learning systems, test samples are often drawn from a different data distribution than the training samples, due to variations in data acquisition processes between the training and test sets – caused by for example, different illumination conditions and cameras in the context of visual tasks. This shift in data distributions, known as domain shift, is a core reason which hinders the generalizability of predictive models to new domains. As manual labeling of data in each test domain is prohibitively expensive, unsupervised domain adaptation **(uDA)** has emerged as a promising solution to transfer the knowledge from a labeled source domain to unlabeled target domains (Long et al. (2015; 2017); Ganin et al. (2016); Tzeng et al. (2017); Hoffman et al. (2018); Shen et al. (2018); Sankaranarayanan et al. (2018); Hoffman et al. (2018)).

While uDA methods are indeed effective, little attention has been paid on how they would be incorporated in real-world machine learning systems. In this paper, we study and propose solutions for practical problems that arise while applying uDA techniques in ML systems, namely the challenges of distributed domain datasets and the overly centralized nature of existing uDA approaches. As a motivating example, consider a scenario wherein a model is trained for the task of fetal head detection from labeled ultrasound images collected in a hospital in Finland (source domain $S_{\text{fin}}$). Subsequently, this pre-trained model has to be deployed in three target hospitals - one in the US ($T_{\text{us1}}$) and two in China ($T_{\text{cn1}}$ and $T_{\text{cn2}}$). Due to variations in sonogram machines and medical training of sonographers, a domain shift is likely to occur in the test samples; hence we need to apply uDA to adapt the source classifier in the target domains.

Existing uDA methods are centralized by design, in that they assume that each target domain would *always* adapt from the labeled source domain ($S_{\text{fin}}$). This raises two issues: firstly, if the machine hosting the labeled source dataset is unavailable (e.g., it is undergoing maintenance or has connectivity issues), then clearly adaptation is not possible. More importantly, we argue that this choice of *always* adapting from a labeled source is not optimal from an adaptation perspective, because the domain discrepancy between the labeled source and a potential target domain could be high in some cases. Our work seeks to explore an interesting proposition: *in addition to adapting from the labeled source, can we also perform uDA with other target domains, which themselves may have undergone domain adaptation in the past.*

Further, existing uDA methods do not support distributed domain datasets and assume that source and target data are available on the same machine. Clearly, this raises privacy and legal concerns since either the source domain or the target domains need to send their sensitive data (i.e., sonograms) to each other to perform uDA. Moreover, such transfer of potentially large datasets also incurs severe communication costs.

In summary, this paper makes the following contributions:

- We formulate and study a brand-new problem focusing on the challenges of uDA in practical machine learning systems.

- We propose a multi-step uDA framework, wherein target domains can adapt not only from the labeled source, but also from other target domains. Powering this framework is a novel collaborator selector algorithm that chooses the optimal adaptation collaborator for each target domain, before initiating the adaptation.

- We propose an effective technique for allowing uDA algorithms and adversarial training to work across distributed datasets. Through extensive experiments on five image and speech datasets, we demonstrate the efficacy of our proposed solution.

## 2 Problem Formulation and Related Work

Consider a practical scenario of deploying a uDA method in a real-world ML system. Assume that a data collection exercise yields a labeled dataset upon which a model is trained using supervised learning. Thereafter, this model needs to be deployed to a population of users (or targets) whose data is unlabeled and divergent from the training dataset. Mathematically, we are presented with a single source domain $S = (\mathbb{R}^n, p_S(x, y))$, with input data $X_S$ and labels $Y_S$. There are multiple target domains $\{T^j | j = 1, \ldots, K\}$, with data $X_T^j$ drawn from target distribution $p_T^j(x, y)$, *without* labeled observations. Using supervised learning, we train a representation mapping, $\mathcal{M}_S$, and a classifier, $\mathcal{C}_S$ for the source domain. However, for the target domains, due to the absence of labeled observations, supervised learning is not possible and hence we would like to do adaptation with the source domain. We assume that target domains are introduced sequentially, one at a time.

Under this problem formulation, we highlight two unexplored research challenges:

**Collaborator Selection.** For each target domain that joins the system, how do we select an optimal collaborator for domain adaptation that will lead to highest post-adaptation accuracy for the target domain? Existing uDA methods (e.g., Hoffman et al. (2018); Tzeng et al. (2017); Ganin et al. (2016) always use the labeled source domain $\mathcal{S}$ as the adaptation collaborator for each target domain, however we argue that it is not optimal to *always* adapt for the labeled source domain for two reasons: i) if the domain shift between the target domain and the labeled source is high, we may not achieve a good adaptation performance (Wulfmeier et al. (2017)); (ii) from a practical perspective, it makes the entire system centralized and prone to failure if the device hosting the source dataset is unavailable for adaptation (e.g., due to connectivity issues). Instead, we propose a multi-step decentralized domain adaptation approach built upon the idea that a new target domain can adapt not only from the labeled source domain, but also from other target domains in the system which have already undergone domain adaptation.

More concretely, we define a collaborator set $\mathcal{C}$ as the set of domains that are available to collaborate with a target domain on an adaptation task. At step $\tau = \tau_0$, only the source domain is present in the system, hence $\mathcal{C}_0 = \{\mathcal{S}\}$. At step $\tau = \tau_1$, the first target domain $T^1$ joins the system – at this moment, only the source domain $S$ has a learned representation $\mathcal{M}_S$. Thereafter, $T^1$ performs uDA with the source $S$ and learns a representation $\mathcal{M}_{T^1}$. Subsequently, we have $\mathcal{C}_1 = \{S, T^1\}$, i.e., future target domains now have two candidates with whom they can collaborate to perform domain adaptation. In general, at step $\tau = \tau_K, \mathcal{C}_K = \{S\} \cup \{T^j | k = 1, \ldots, K\}$.

In §3.3, we propose an algorithm to select an optimal collaborator $c \in \mathcal{C}_K$ for each target domain, that is likely to yield the best accuracy post-adaptation. Once the optimal collaborator is chosen, any existing uDA method could be employed to perform the pairwise adaptation between the chosen collaborator and the target domain. In §4, we show the effectiveness of this decentralized and multi-step adaptation approach against conventional baselines and across a number of uDA techniques.

**Distributed and Private Data**. Existing uDA methods assume that the datasets from the source domain $\mathcal{S}$ and a given target domain $T^j$ are available on the same compute unit (e.g., on a server). However, this assumption is violated in practical scenarios, as the datasets are often private, and users or companies may not be allowed to share them with other parties due to legal reasons such as the data privacy law (GDPR) in Europe. In addition to the privacy issue, exchanging potentially large datasets incur high communication costs, making it undesirable in practical settings. This raises the question: can we make domain adaptation methods to work in a distributed and privacy-preserving manner such that the collaborators in the domain adaptation process can keep their data private and still receive the benefits of adaptation?

These challenges about making uDA algorithms work in a decentralized and distributed way are closely related, and addressing them is critical for practical usage of uDA methods. We first describe (in §3.2) our idea of supporting uDA with distributed domain datasets, by enabling knowledge exchange between adaptation collaborators using the gradients of Discriminator, which allows for the raw data and features of each domain to remain private. Next, in §3.3, we propose an algorithm for selecting an optimal collaborator for each target domain.

## 2.1 Related Work

Bobu et al. (2018) presented the problem of continuous unsupervised adaptation wherein the target domain is smoothly varying temporally. They proposed an iterative uDA method with a replay loss to prevent the model from forgetting knowledge from past domains. Although our solution of multi-step domain adaptation is also iterative, we do not assume any smooth ordering between target domains, hence warranting the need for collaboration selection. Zhao et al. (2018) proposed MDAN where a target domain can adapt from multiple labeled source domains. As there is only one labeled source domain in our setup, this method cannot be directly applied. However, we can still combine the adversarial losses from multiple domains as proposed in MDAN - as such, we implement a variant of MDAN as a baseline for our method.

In our paper, the training data from different domains is distributed in separate nodes, which relates our problem to distributed training. The motivation of most existing works in this field is to accelerate the training process by partitioning the training data into multiple computational nodes and calculating the gradients in parallel. If the results from all nodes are aggregated globally by a central node (e.g., parameter server (Li et al. (2014); Dai et al. (2015))), or by message passing process (e.g., All-Reduce (Sergeev & Del Balso (2018))), then it is called centralized distributed training. By contrast, if the system does not require to synchronize globally (Lian et al. (2017a); Tang et al. (2018b)), it is called decentralized training. Besides, decentralized training can be further accelerated by its conjunction with low-precision training (Tang et al. (2018a)), or asynchronous training (Lian et al. (2017b)). In addition to acceleration, federated learning (Bonawitz et al. (2019); Konečný et al. (2016); Yang et al. (2019); Liu et al. (2018)) considers a training setting where the training data is stored over a large number of separate devices and cannot be shared due to privacy reasons. Our work builds upon the distributed training literature and we show, for the first time, how distributed training can be used in the context of uDA. Further, it is important to clarify that we use 'decentralization' in this paper in the context of domain adaptation, i.e., target domains can choose to adapt from any one of the available domains, instead of a single source domain (akin to a central node). This makes our techniques and their subsequent evaluation different from traditional decentralized learning papers.

Finally, an important goal of our paper is to support uDA while keeping the domain datasets private. There has been active research recently at the intersection of privacy and ML. Melis et al. (2019) showed that collaborative learning is vulnerable to membership inference attacks and leakage of dataset properties, and suggested employing dimensionality reduction and Dropout as ways to mitigate these attacks. Aono et al. (2017) demonstrated that the gradients of a classifier exchanged in distributed training can be used to reconstruct the raw training data and proposed using additively homomorphic encryption to fix the problem. Geyer et al. (2017) proposed a differential privacy approach to hide a single client's data contribution during federated learning. Finally, Nasr et al. (2018) did an in-depth analysis of passive and active white-box attacks on deep learning models. To the best of our knowledge, no prior work has shown that sharing gradients of a discriminator in adversarial learning – which is the key component of our approach – can leak raw training data. As such, our approach provides clear privacy benefits over conventional uDA techniques. That said, we do not discount the possibility that privacy attacks could be developed on our technique in the future, and leave it open as a topic of further research.

# 3 Our Approach

## 3.1 Primer on Adversarial Domain Adaptation

Before describing our approach, we provide a primer on adversarial domain adaptation which is a widely used approach for uDA and serves as the basic foundation of our solution. The core idea here is to use adversarial learning to align the feature representations of the source and target domains, thereby allowing a source classifier to be used in the target domain. Initially, a source extractor $E_S$ and a source classifier $C_S$ are trained using supervised learning by solving the optimization problem:

$$\min_{E_S, C_S} \mathcal{L}_{cls} = -\mathbb{E}_{(x_s, y_s) \sim (X_S, Y_S)} \sum_{k=1}^{K} \mathbb{1}_{[k=y_s]} [\log(C_S(E_S(x_s)))]$$

The goal of the adaptation process is to learn a target extractor $E_T$ using adversarial learning. To this end, the source extractor $E_S$ is used to initialize the target extractor $E_T$. The weights of the source model are fixed during adversarial training. As in standard adversarial learning, two losses are optimized in the training process, a discriminator loss $\mathcal{L}_{adv_D}$ and a mapping loss $\mathcal{L}_{adv_M}$. Different uDA methods compute these two losses in their own way, e.g., ADDA (Tzeng et al. (2017)) uses the following loss formulations:

$$\min_D \mathcal{L}_{adv_D} = -\mathbb{E}_{x_s \sim \mathcal{X}_S}[\log(D(E_S(x_s)))] - \mathbb{E}_{x_t \sim \mathcal{X}_T}[\log(1 - D(E_T(x_t)))] \tag{1}$$

$$\min_{E_T} \mathcal{L}_{adv_M} = -\mathbb{E}_{x_t \sim \mathcal{X}_T}[\log(D(E_T(x_t)))] \tag{2}$$

where $D$ represents a domain discriminator which aims to distinguish source domain data and target domain data. Details about other uDA methods are provided in §A.4 in the Appendix.

Our work in general applies to any feature-alignment based uDA technique, which allows for source and target encoders to be trained independently. We note that there are indeed uDA approaches which optimize the source classification loss and adversarial loss in the same batch during training, but they are not compatible with our problem formulation wherein a pre-trained source model is to be deployed on a set of target domains. Further, uDA approaches based on generative modeling (Hoffman et al. (2018); Hosseini-Asl et al. (2018)) are also out of scope of our work.

## 3.2 Distributed Domain Adaptation

Prior adversarial uDA methods assume that source and target data is available on the same compute unit to perform adversarial learning. However as discussed in §2, data from different domains is often distributed and private in realistic applications, and sharing it could be legally prohibited. We present an approach of performing domain adaptation with distributed domain datasets. In this vein, two key questions are: a) how to distribute the adversarial network architecture across nodes?, b) how to exchange information between the source and target nodes during training?

We propose to split each constituent neural network in the adversarial training framework across source and target nodes. Consequently, the *extractor*, *domain discriminator* and *task classifier* have source components ($E_S$, $D_S$, $C_S$) and target components ($E_T$, $D_T$, $C_T$). Source data and target data can therefore be fed separately into their own model components to prevent any exchange of raw data or feature representations across nodes. Given this network architecture, we now present our training strategy (also summarized in Algorithm 1). The first step is pre-training and initialization. We assume that the extractor and classifier ($E_S$ and $C_S$) of the source domain have been trained either using supervised learning or through prior domain adaptation, and the source discriminator $D_S$ is initialized randomly. Thereafter, the target domain components, $E_T$, $C_T$ and $D_T$, are initialized with the respective source components. Figure 3 in Appendix A illustrates our distributed architecture and in §A.4, we provide the equations for various adversarial losses.

During adversarial training, for each step, source and target domains sample a training batch from their own domain data and calculate stochastic gradients for $D_S$ on source domain data and for $E_T$, $D_T$ on target domain data accordingly. However, as shown in Eq.1, the discriminator needs to be trained on data from both the source and target domains, therefore we need a mechanism to transfer knowledge across between source and target domains. As opposed to transferring the raw data or extracted features across nodes, our approach performs knowledge exchange through the *gradients of the discriminators*. This in turn does afford certain privacy benefits, as we do not transmit raw data or the extracted features or even the gradients of the feature extractor which can potentially leak raw data (Aono et al. (2017)). Although to the best of our knowledge, no prior work has shown that discriminator gradients can leak training data, we do not discount the possibility that such privacy attacks can be developed in the future. We leave their study and mitigation as a future work.

A simple strategy for knowledge transfer between nodes would be to exchange and average the gradients of the two discriminators after each step, and then update $D_S$ and $D_T$ with the averaged gradients. While this will ensure that the discriminators are synchronized, it incurs a significant communication cost for each step. Instead, we propose a simple but effective method, called *Lazy Synchronization*, to reduce the communication cost of the algorithm. The basic idea is to synchronize the source and target discriminators every $p$ training steps. We refer to the training steps at which the synchronization takes place as the *sync-up steps* while the other steps during which both nodes are computing the gradients locally are called *local steps*. For discriminators $D_S$ and $D_T$, their local

gradients are accumulated during *local steps*, and during the *sync-up steps* the accumulated gradients are averaged and applied to $D_S$ and $D_T$. Through this, we can ensure there is no divergence between the discriminators, and at the same time we are able to decrease the communication cost to $1/p$. Note that as we only exchange discriminator gradients, our communication costs are independent of the feature extractors, which can be potentially very deep and large in size.

---

**Algorithm 1: DISTRIBUTED_UDA: Lazy-Synchronized Distributed Domain Adaptation**

**Result:** $E_T$ and $C_T$

1 **Input**: Pre-trained $E_S$ and $C_S$; Randomly Initialize $D_S$; Initialize $E_T = E_S$, $D_T = D_S$, and $C_T = C_S$; Sync up step $p$; number of total steps $N$;

2 **for** $n = 1, 2, ..., N$ **do**

3      Source node and target node sample a batch of data respectively, $\xi_s^{(n)}$ and $\xi_t^{(n)}$;

4      Calculate gradients $\nabla g(D_S, \xi_s^{(n)})$ on source node, gradients $\nabla g(E_T, \xi_t^{(n)})$ and $\nabla g(D_T, \xi_t^{(n)})$ on target node ;

5      **if** *isTargetNode* **then**

6          Apply $\nabla g(E_T, \xi_t^{(n)})$ to $E_T$

7      **end**

8      Add $\nabla g(D_S, \xi_s^{(n)})$ to source gradients buffer $G_S$, add $\nabla g(D_T, \xi_t^{(n)})$ to target gradients buffer $G_T$;

9      **if** $n \% p == 0$ **then**

10          Exchange gradients buffer and calculate averaged gradients $g_{avg} = \frac{G_S + G_T}{2p}$ ;

11          Apply $g_{avg}$ to $D_S$ and $D_T$ ;

12          Clear $G_S$ and $G_T$ ;

13      **end**

14 **end**

---

**Algorithm 2: COLAB_SELECT: Wasserstein-Distance Guided Collaborator Selection**

1 **Input**: Candidate set $\mathbb{C} = \{(p_1, E_1, C_1), (p_2, E_2, C_2) \ldots (p_N, E_N, C_N)\}$, Target Dataset $p_T$;

2 **for** *$(p_i, E_i, C_i)$ in $\mathbb{C}$* **do**

3      $W_i$ = compute_wasserstein_distance($p_i$, $p_T$);

4      Compute source error $\epsilon_s$ using a small labeled test set.;

5      Compute the Lipschitz Constant $K$ for the source extractor $E_i$ and classifier $C_i$. ;

6      $\epsilon_{t_{max}}^i = \epsilon_s + 2.K.W_i$;

7 **end**

8 Optimal Collaborator $O = \underset{i=1...N}{\arg\min}$ $(\epsilon_{t_{max}}^i)$;

9 Return $E_O$ and $C_O$;

---

**Algorithm 3: MDDA: Multi-step Decentralized Domain Selection**

1 **Input**: Initial Candidate set $\mathbb{C} = \{(E_S, C_S)\}$, Ordering of $K$ target domains $\mathbb{T} = T_1, T_2 \ldots T_K$;

2 **for** *$T_i$ in $\mathbb{T}$* **do**

3      Obtain the optimal collaborator $O \in \mathbb{C}$ using COLAB_SELECT;

4      Run DISTRIBUTED_DA between $O$ and $T_i$ to obtain $E_{T_i}$ and $C_{T_i}$ ;

5      $\mathbb{C} \leftarrow \mathbb{C} \cup \{(E_{T_i}, C_{T_i})\}$;

6 **end**

---

The target extractor $E_T$ is independent of the source data, therefore it is updated for each batch using the adversarial formulation proposed by the underlying adaptation algorithm. For instance, Equation 9 shows the loss function to update $E_T$ as proposed in ADDA. On the other hand, GRL (Ganin et al. (2016)) uses a gradient reversal technique which results in $\mathcal{L}_{adv_M} = -L_{adv_D}$, Shen et al. (2018) uses the Wasserstein Distance as a metric for adversarial training. Details about the adversarial loss functions of different algorithms used in this paper are provided in §A.4. Finally, the target classifier $C_T$ is initialized with $C_S$ and is not updated in the training process. The intuition is that if the feature space of the target domain can be successfully aligned with the source domain, then $C_S$ can directly be used in the target domain without any adaptation.

In §4.2, we empirically show that our method can obtain comparable accuracy to non-distributed training algorithms, while at the same time, preserving private user data and minimizing the communication costs in the training process.

### 3.3 Wasserstein distance guided collaborator selection

We now discuss how to select an optimal adaptation collaborator $c \in \mathcal{C}_K$ for each target domain. Indeed, we should select a collaborator that is likely to yield the lowest target error post adaptation, but can we choose the optimal collaborator even before performing domain adaptation? In a

seminal theoretical work, Ben-David et al. (2010) showed that the target error is bounded by the sum of source error and the divergence between source and target distributions. Redko et al. (2017) and Shen et al. (2018) did a similar analysis using the Wasserstein distance, which, together with Lipschitz functions, forms the basis for our collaborator selection method.

A function $f : X \to \mathbb{R}$ is $\theta$-Lipschitz if it satisfies the inequality $\|f(x) - f(y)\| \leq \theta \|x - y\|$ for some $\theta \in \mathbb{R}_+$. The smallest such $\theta$ is called the Lipschitz constant $\mathrm{Lip}(f)$ of $f$. Further, the 1-Wasserstein distance (Villani (2008)) between two distributions $p_1$ and $p_2$ – using the duality formula – is

$$W_1(p_1, p_2) = \sup_{f : \mathrm{Lip}(f) \leq 1} \mathbb{E}_{p_1}[f] - \mathbb{E}_{p_2}[f] \tag{3}$$

Shen et al. (2018) computed the following robustness bound

$$\epsilon_{p_2}(h, h') \leq \epsilon_{p_1}(h, h') + 2\theta W_1(p_1, p_2) \tag{4}$$

for any two $\theta$-Lipschitz hypotheses $h$ and $h'$, where we denoted

$$\epsilon_p(h, h') = \mathbb{E}_p[\|h - h'\|]. \tag{5}$$

Our method is motivated by the observation that the above bound is a good approximation of the error $\epsilon_{p_2}(h) = \epsilon_{p_2}(h, h^{\mathrm{true}})$, provided that both the true hypothesis $h^{\mathrm{true}}$ and the learned hypotheses $h = E \circ C$ are $\theta$-Lipschitz, and the push-forward measures $E_* p$ are close together in the Wasserstein distance, for a sufficiently small $\theta$. These conditions are motivated by the fact that the domains should be well aligned for unsupervised domain adaptation to be effective. Using a Lipschitz continuous extractor allows us to perform adversarial alignment of higher order features, while maintaining a Lipschitz continuity guarantee on the true hypothesis as a function of the encoded features $E(x)$, where $x \sim p_2$.

Now assume we are trying to use domain adaptation to find a model for a domain $D = (\mathbb{R}^n, p)$ and we have a set of *candidate* domains $D_k = (\mathbb{R}^n, p_k)$, $k = 1, \ldots, K$ each with a pre-trained model $\mathcal{M}_k = (E_k, C_k)$, obtained by using either supervised learning or prior domain adaptation. We use the estimate provided by Equation 4 to select the optimal collaborator domain

$$D_{opt} = \operatorname*{argmin}_{k=1,\ldots,K} \epsilon_{p_k}(E_k \circ C_k) + 2\theta W_1(p_k, p) \tag{6}$$

and thereafter perform adaptation from $D_{\mathrm{opt}}$ to $D$.

**Enforcing Lipschitz continuity and computing the Wasserstein distance**. Training neural networks with minimal Lipschitz constants can be done by directly regularizing the spectral norm of the linear part of each layer Mises & Pollaczek-Geiringer (1929); Gouk et al. (2018), or a gradient penalty Gulrajani et al. (2017), or indirectly by using $L1$ or $L2$ weight decay. Enforcing a hard constraint on $\theta$ can be done by rescaling the linear part of each layer or, indirectly, by weight clipping. For the feature extractor and classifier neural networks, we employ Spectral Norm regularization proposed by Gouk et al. (2018). Further, to compute the Wasserstein distance between distributions, we train a Wasserstein critic with gradient penalty as proposed by Gulrajani et al. (2017). This critic can also be trained in a distributed manner following our lazy synchronization strategy (§ 3.2).

### 3.4 Multi-Step Decentralized Domain Adaptation (MDDA)

Using the two core ideas discussed § 3.2 and § 3.3, we now summarize our multi-step DA algorithm MDDA (Algorithm 3). Assume we are given a source domain with pre-trained model and an arbitrary ordering of $K$ unlabeled target domains. For every target domain that joins the system, we run collaborator selection with the candidates available in the system. Upon selecting an optimal collaborator, we run the distributed DA algorithm to enable uDA in a privacy-preserving manner. The adaptation process results in a feature extractor $E_T$ and a classifier $C_T$ for the target domain. Finally, the recently adapted target domain is added to the candidate set, and may serve as a potential collaborator for future domains.

## 4 Evaluation

### 4.1 Implementation

**Datasets**. We conduct experiments on five image and audio datasets: Rotated MNIST, Digits, and Office-Caltech, DomainNet and Mic2Mic. *Rotated MNIST* is a variant of MNIST with numbers rotated from 0°to 330°at increments of 30°. Each rotation is considered a separate domain. The

*Digits* adaptation task has five domains: MNIST (M), USPS (U), SVHN (S) (Netzer et al. (2011)), MNIST-M and SynNumbers (SYN) (Ganin & Lempitsky (2014)). Each domain consists of 10 digit classes ranging from 0-9 in different styles. The *Office-Caltech* dataset contains object images from 10 classes obtained from Amazon (A), DSLR camera (D), Web camera (W), and Caltech-256 (C). *DomainNet* (Peng et al. (2018)) is a new dataset from which we use four labeled image domains containing 345 classes each: Real (R), QuickDraw (Q), Infograph (I), and Sketch (S). Finally, Mic2Mic (Mathur et al. (2019)) is a speech-based keyword detection dataset wherein the keywords are recorded with four different microphones: Matrix Creator (C), Matrix Voice (V), ReSpeaker (R) and USB (U). Each microphone represents a domain.

**Baselines.** To evaluate our collaborator selection strategy, we compare it against the following:

- **Labeled Source.** This represents the most commonly used approach in uDA works, wherein every target domain performs a pairwise adaptation with the labeled source domain $S$.
- **Random Collaborator.** Here, we randomly choose a collaborator $c \in \mathcal{C}_K$ from the collaborator set and perform adaptation with the target domain.
- **Multi-Collaborator.** This approach is based on MDAN proposed by Zhao et al. (2018), wherein all available candidate domains are used for adaptation, however the contribution of each domain is dynamically re-weighted in the adversarial training process. However, note that MDAN assumes that all the candidate domains are labeled and jointly optimize the classification loss with the adversarial training. In our scenario, only one candidate domain has labeled data ($S$) while others are unlabeled. As such, we implement a modified version of MDAN which only optimizes their proposed adversarial loss in line with our problem formulation in §3.1.

Further, we compare our proposed distributed adversarial learning architecture against (i) a non-distributed uDA baseline wherein the source and target data reside on the same node, and (ii) D-PSGD proposed by Lian et al. (2017a) for distributed training of neural networks.

**Experiment Setup.** We follow the same evaluation protocol as earlier uDA works (Hoffman et al. (2018); Tzeng et al. (2017)) wherein the unlabeled training instances from source and target domains are used for adversarial training, and the adapted model is evaluated on a held-out test dataset from target domain. In addition, we use a small subset of the training instances (10% of the total instances) for doing collaborator selection. Further, as we explained earlier, we assume that target domains join the system sequentially in an order, one at a time. Therefore, for each dataset, we randomly select different orderings of source and target domains and present average results across them. Our system is implemented with Tensorflow 2.0 and trained on Nvidia V100 GPUs. For more details about network architectures and training hyper-parameters, please see Appendix A.

## 4.2 Results

We now present our results of applying MDDA on five datasets. Overall, our results show that MDDA selects the right collaborator for adaptation in 82% of the cases, which result in the highest mean target domain accuracy when compared to other baselines.

**Performance of Collaborator Selection.** For each dataset, we choose three random orderings of source and target domains, e.g., for *Office-Caltech*, we choose $O_1$=**W**,A,D,C; $O_2$=**D**,C,A,W; and $O_3$=**A**,D,W,C. Here the first domain in the order (in bold) represents the labeled source domain, while the others are unlabeled target domains in the order in which they join the system. Please refer to Appendix A for details about the orderings used for other datasets.

Figure 1 shows the accuracy of our collaborator selection algorithm. For this experiment, we perform adaptation between a target domain and each available candidate, and based on the target accuracy after each pairwise adaptation, we obtain the collaborator that yields the highest accuracy for each target domain. This serves as the ground truth for our algorithm. Thereafter, we run our collaborator selection algorithm on each ordering and obtain the optimal collaborator based on Algorithm 2. By comparing the output of our algorithm with the ground truth, we obtain Figure 1 which shows that on average, our algorithm has a selection accuracy of 82%. On further analysis, we found that our algorithm primarily makes mistakes when two candidates have similar domain discrepancy (estimated using the $W_1$ distance) with the target domain and we pick the second most optimal collaborator. In such scenarios, although the top-1 collaborator selection accuracy drops,

it typically does not impact the target error significantly because adapting with the second-most optimal domain also yields a good adaptation performance.

| | RMNIST | | | Mic2Mic | | | Digits | | | Office-Caltech | | | DomainNet | |
| --- | --- | --- | --- | --- | --- | --- | --- | --- | --- | --- | --- | --- | --- | --- |
| | $O_1$ | $O_2$ | $O_3$ | $O_1$ | $O_2$ | $O_3$ | $O_1$ | $O_2$ | $O_3$ | $O_1$ | $O_2$ | $O_3$ | $O_1$ | $O_2$ |
| No Adaptation | 36.23 | 36.23 | 35.80 | 73.11 | 73.38 | 74.7 | 64.34 | 73.42 | 57.80 | 93.41 | 87.15 | 92.32 | 19.09 | 19.03 |
| Random | 42.89 | 39.73 | 43.49 | 76.9 | 79.1 | 78.0 | 65.11 | 78.41 | 59.5 | 95.45 | 90.10 | 95.69 | 32.21 | 33.19 |
| Labeled Source | 53.62 | 54.53 | 46.47 | **78.07** | **80.96** | **79.96** | 66.02 | 81.46 | **65.89** | 95.45 | **91.98** | 95.57 | **34.28** | 34.21 |
| Multi-Collaborator | 36.08 | 37.88 | 37.84 | 74.34 | 75.19 | 75.37 | 60.63 | 64.19 | 62.91 | 87.64 | 88.33 | 92.37 | 25.3 | 24.1 |
| MDDA (Ours) | **86.08** | **70.84** | **79.19** | **78.07** | 80.12 | **79.96** | **81.87** | **89.03** | 64.85 | **95.82** | 89.58 | **96.01** | 34.21 | **34.34** |
| Ideal | 86.38 | 71.77 | 79.52 | 78.07 | 80.96 | 79.96 | 83.84 | 89.03 | 69.32 | 95.82 | 92.09 | 96.01 | 34.28 | 34.34 |

Table 1: Target accuracy under different techniques of selecting a collaborator using ADDA. *Ideal* refers to the best achievable performance if the most optimal collaborator is picked for each target.

Table 1 shows, for three different orderings, the mean accuracy across all target domains with MDDA and other baselines. We observe that in majority of the scenarios, MDDA outperforms the labeled-source adaptation baseline which is commonly used in uDA methods. This confirms our key intuition that a labeled source domain is not always optimal for domain adaptation, and demonstrates the value of a more flexible and decentralized approach like MDDA. Interestingly, we observe that while the multi-collaborator baseline is less accurate than MDDA in general, its performance is particularly poor on Rotated MNIST in which the number of collaborators are quite high – while a detailed investigation of this finding is outside the scope of this paper, we surmise that this technique does not scale well as the number of collaborator domains increase. MDDA, on the contrary, only performs pairwise adaptation once an optimal collaborator is selected, as such it scales gracefully as the number of target domains increase.

The results presented so far used the adversarial loss formulation from Eq. 1 and 2 which was proposed in ADDA. In Table 2, we evaluate the applicability of MDDA to methods that use other adversarial loss formulations such as (i) when a Gradient Reversal Layer (GRL) is used to compute the mapping loss, (ii) when Wasserstein Distance is used as a loss metric for domain discriminator (Shen et al. (2018)) and (iii) a recently proposed uDA technique called CADA (Zou et al. (2019)) which enforces consensus between source and target features. More details about these techniques and their optimization objectives are provided in Appendix A.

We observe that while different techniques yield different target accuracies, MDDA can work in conjunction with all of them to improve the overall accuracy over the conventional labeled source baseline.

| | RMNIST ($O_1$) | | | | Digits ($O_2$) | | | |
| --- | --- | --- | --- | --- | --- | --- | --- | --- |
| | ADDA | GRL | WassDA | CADA | ADDA | GRL | WassDA | CADA |
| No Adaptation | 36.23 | 36.23 | 36.23 | 36.23 | 73.42 | 73.42 | 73.42 | 73.42 |
| Random | 42.89 | 39.33 | 43.78 | 57.13 | 78.41 | 79.22 | 74.91 | 76.70 |
| Labeled Source | 53.62 | 45.60 | 37.99 | 37.65 | 81.46 | 76.66 | 77.31 | 65.22 |
| MDDA(Ours) | **86.08** | **63.49** | **85.6** | **78.37** | **89.03** | **85.24** | **79.48** | **76.79** |
| Ideal | 86.38 | 68.92 | 86.94 | 79.50 | 89.92 | 85.42 | 79.9 | 78.1 |

Table 2: Mean target accuracy for a random order of domains for different adaptation methods.

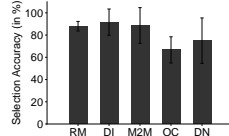

Figure 1: Mean accuracy of collaborator selection for three orderings for Rotated MNIST (RM), Digits (DI), Mic2Mic (M2M), Office (OC) and DomainNet(DN).

**Performance of Distributed Domain Adaptation.** We now evaluate our Lazy-synchronized DA algorithm from a domain adaptation lens. Our key objective is to evaluate the target-domain accuracy and communication trade-off, i.e., can we save communication costs associated with distributed training while ensuring similar levels of target domain accuracy as the conventional non-distributed DA algorithms. In Figure 2, we plot the target domain accuracy with the number of adversarial training steps for three uDA tasks. It can be observed that even with lazy synchronization (sync-up step size $p = 4$), we can achieve similar target accuracy and convergence rate as non-distributed and synchronized training methods, while incurring only 25% of the communication costs (as we are exchanging data every 4 steps). We also note that the distributed training baseline (DPSGD) performs poorly for adversarial training primarily because of the non-overlapping label spaces on each node (with respect to the domain label).

Table 3 expands this finding to show the target domain accuracy for 10 adaptation tasks under different training mechanisms. Further, we also report the average time taken to perform the adversarial training for different datasets. Note that the training time is the sum of local computation time and

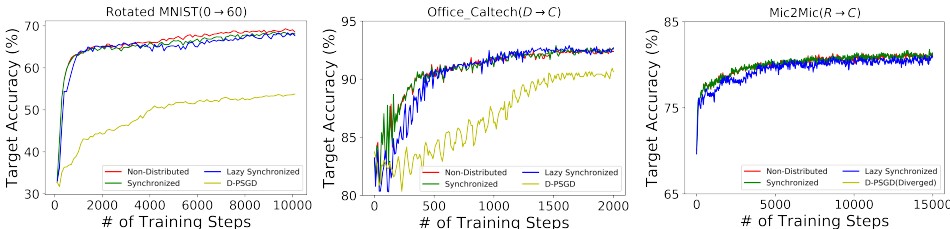

Figure 2: Comparison of target domain accuracy and training convergence across non-distributed and distributed training methods. For Lazy-synchronized approach, we use sync-up step $p = 4$. For Mic2Mic, the PSGD baseline did not improve the accuracy, as such we omit it from the figure.

communication time across nodes. In conventional non-distributed DA, it is assumed that the domain datasets are available on the same node, as such there is no communication required during the training, which expectedly results in the fastest training process, at the expense of user privacy. In the fully-synchronized case, the amount of communication equals the size of discriminator gradients multiplied by the number of training steps, whereas in the lazy-synchronized case, the communication amount is one $p$th of the amount of fully synchronized case, where $p$ is the sync-up step. Based on the amount of data communicated across nodes, we estimate the communication time by assuming a bandwidth of 40Mbps which is roughly the average upload speed for broadband connections globally[1]. The computation time is measured for each training step and aggregated to get the total computation time. By adding the total computation time with the total communication time, we obtain the total training time reported in Table 3.

As shown in Table 3, the Lazy-Synchronized approach achieves target accuracy close to non-distributed and fully-synchronized approaches. For example, in the D $\rightarrow$ A task in Office-Caltech, our approach – while ensuring that domain data remains private – results in 91.77% target domain accuracy, which is indeed close to the 92.01% and 92.13% accuracy of non-distributed and fully-synchronized training mechanism. Further, we observe that the training time of our approach is significantly less than the fully-synchronized approach due to the reduced communication.

| Training | RMNIST | | | Office-Caltech | | | Digits | | | DomainNet | | | Mic2Mic | | |
|---|---|---|---|---|---|---|---|---|---|---|---|---|---|---|---|
| | $0 \rightarrow 60$ | $150 \rightarrow 180$ | t (min) | W $\rightarrow$ C | D $\rightarrow$ A | t (min) | M-M $\rightarrow$ U | Syn $\rightarrow$ U | t (min) | S $\rightarrow$ R | Q $\rightarrow$ R | t (min) | C $\rightarrow$ R | R $\rightarrow$ C | t (min) |
| Source-Only | 32.68 | 61.51 | - | 87.81 | 85.28 | - | 57.54 | 79.32 | - | 34.34 | 30.75 | - | 69.65 | 69.11 | - |
| Non-Distributed | 69.61 | 91.86 | 0.15 | 91.74 | 92.01 | 11 | 82.1 | 90.1 | 0.67 | 54.91 | 50.67 | 1.12 | 78.02 | 77.94 | 5.8 |
| Synchronized | 69.30 | 91.21 | 6.8 | 91.02 | 92.13 | 28 | 82.13 | 89.9 | 20.7 | 54.23 | 49.95 | 17.5 | 81.5 | 81.1 | 22.5 |
| Lazy Synchronized | 68.34 | 90.16 | 1.8 | 90.56 | 91.77 | 15.3 | 81.66 | 89.78 | 5.7 | 53.25 | 49.02 | 5.2 | 77.48 | 81.03 | 10 |

Table 3: Target Domain Accuracy and Training Time (averaged across two adaptation tasks) for each dataset. As expected, the *Non-Distributed* approach is the fastest as it does not require any communication during training, at the expense of user privacy. Our Lazy-Synchronized approach ($p$=4) provides the best trade-off between accuracy and training time, without requiring the exchange of raw domain data.

# 5 Limitations and Conclusion

We introduced a novel perspective on uDA research and explored practical challenges associated with deploying uDA methods in real-world ML systems. Our proposed framework MDDA is the first-ever solution aiming to make uDA work in a decentralized and distributed manner.

As uDA is a rapidly evolving field, we did not study every class of uDA algorithms (e.g., those which combine feature-level adaptations with instance-level adaptations) in this paper. We also made an assumption that target domains are introduced sequentially in the system, however there could indeed be other ways in which ML models would evolve in practice (e.g., multiple target domains join together or in batches). We leave those scenarios as future work. Finally, our method provides user privacy by only sending discriminator gradients across nodes. While no prior work has shown that discriminator gradients can leak raw data, we do not discount the possibility that privacy attacks could be developed on MDDA in the future. We leave a detailed privacy study of MDDA as a future work.

---

[1]https://www.speedtest.net/global-index

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

# A    Appendix

Here we provide details about our experiment setup and model architectures. We are also in the process of obtaining necessary clearances to release our source code to the community.

## A.1    Domain Orderings.

As we mentioned in our problem formulation, the order of target domains joining the system can effect the decisions of our collaborator selecting algorithm, and lead to different target accuracies. To measure the effect of target orders, we reported three different orders for each data set in our experiments. We specify these order as follows:

| | $O_1$ | $O_2$ | $O_3$ |
|---|---|---|---|
| RMNIST | **0**,30,60,90,120,150, 180,210,240,270,300,330 | **0**,180,210,240,270,300, 330,30,60,90,120,150 | **30**,0,180,210,240,330, 60,90,270,300,120,150 |
| Mic2Mic | **C**,V,R,U | **R**,U,C,V | **V**,R,U,C |
| Digits | **svhn**,mnist,mnist_modified, synth_digits,usps | **synth_digits**,usps,mnist, mnist_modified,svhn | **mnist_modified**,mnist,svhn, usps,synth_digits |
| Office-Caltech | **W**,A,D,C | **D**,C,A,W | **A**,D,W,C |
| DomainNet | **S**,R,I,Q | **S**,Q,I,R | **R**,I,Q,S |

Table 4: Ordering of source and target used in our experiments. Domains in bold correspond to the labeled source domain, which is introduced first in the system.

## A.2    Model Architectures and Hyperparameters.

We now describe the neural architectures used for each dataset along with the hyperparameters used in supervised and adversarial learning.

**Rotated MNIST:** We use the well-known LeNet architecture for this dataset as shown below. The model was trained for each source domain with a learning rate of $10^{-4}$ using the Adam optimizer and a batch size of 32.

```
Conv2D(filters = 20, kernel_size = 5, activation='relu'),
MaxPooling2D(pool_size = 2, strides = 2),
Conv2D(filters = 50, kernel_size = 5, activation='relu'),
MaxPooling2D(pool_size = 2, strides = 2),
Flatten(),
Dense(500, activation='relu'),
Dense(10, activation='softmax')
```

In order to enforce the Lipschitz continuity, we added spectral norm regularization during the training process. In the adversarial training process, we used the ADDA losses to perform domain adaptation with a learning rate of $10^{-5}$ for the target extractor and $10^{-4}$ for the discriminator.

**Office-Caltech:** We used Keras Inception-V3 pre-trained on ImageNet as the base model for this task. We added a bottleneck layer and a final classification layer. The model was trained for each source domain with a learning rate of $10^{-5}$ using the Adam optimizer and a batch size of 32.

```
InceptionV3(include_top=False, input_shape=(299, 299, 3),avg='pool'),
Dense(256, activation='relu')
Dense(10, activation='softmax')
```

In order to enforce the Lipschitz continuity, we added spectral norm regularization during the training process. In the adversarial training process, we used the ADDA losses to perform domain adaptation with a learning rate of $10^{-5}$ for the target extractor and 1e-4 for the discriminator.

**DomainNet:** We used Keras ResNet50-v2 pre-trained on ImageNet as the base model for this task. The model was trained for each source domain with a learning rate of 1e-5 using the Adam optimizer and a batch size of 64.

```
ResNet50V2(include_top=False, input_shape=(224, 224,3), avg='pool'),
Dense(245, activation='softmax')
```

In order to enforce the Lipschitz continuity, we added spectral norm regularization during the training process. In the adversarial training process, we used the ADDA losses to perform domain adaptation with a learning rate of $10^{-6}$ for the target extractor and $10^{-4}$ for the discriminator.

**Mic2Mic:** We mainly used three convolutional layers for this task. The model was trained for each source domain with a learning rate of $10^{-5}$ using the Adam optimizer and a batch size of 64.

```
Conv2D(filters = 64, kernel_size = (8,20), activation='relu')
MaxPooling2D(pool_size = (2,2)),
Conv2D(filters = 128, kernel_size = (4,10), activation='relu'),
MaxPooling2D(pool_size = (1,4)),
Conv2D(filters = 512, kernel_size = (2,2)),
Flatten(),
Dense(256, activation='relu'),
Dense(31)
```

In order to enforce the Lipschitz continuity, we added spectral norm regularization during the training process. In the adversarial training process, we used the ADDA losses to perform domain adaptation with a learning rate of $10^{-6}$ for the target extractor and $10^{-4}$ for the discriminator.

**Digits:** We constructed a neural network with three convolutional layer and some additional techniques like Dropout, BatchNormalization, for this task. The model was trained for each source domain with a learning rate of $10^{-5}$ using the Adam optimizer and a batch size of 64.

```
inputs = tf.keras.Input(shape=(32,32,3), name='img')
x = Conv2D(filters = 64, kernel_size = 5, strides=2)(inputs)
x = BatchNormalization()(x, training=is_training)
x = Dropout(0.1)(x, training=is_training)
x = ReLU()(x)
x = Conv2D(filters = 128, kernel_size = 5, strides=1)(x)
x = BatchNormalization()(x, training=is_training)
x = Dropout(0.3)(x, training=is_training)
x = ReLU()(x)
x = Conv2D(filters = 256, kernel_size = 5, strides=1)(x)
x = BatchNormalization()(x, training=is_training)
x = Dropout(0.5)(x, training=is_training)
x = ReLU()(x)
x = Flatten()(x)
x = Dense(512)(x)
x = BatchNormalization()(x, training=is_training)
x = ReLU()(x)
x = Dropout(0.5)(x, training=is_training)
```

```
outputs = Dense(10)(x)
```

In order to enforce the Lipschitz continuity, we added spectral norm regularization during the training process. In the adversarial training process, we used the ADDA losses to perform domain adaptation with a learning rate of $10^{-6}$ for the target extractor and $10^{-4}$ for the discriminator.

## A.3 Architecture Diagram

Figure 3 shows the general architecture of our distributed lazy-synchronized training technique. As explained in Algorithm 1 and § 3.2, we split each constituent neural network in the adversarial training framework across source and target nodes. Consequently, the *extractor*, *domain discriminator* and *task classifier* have source components and target components. Knowledge exchange between the nodes is done through exchanging the discriminator gradients in the sync-up steps.

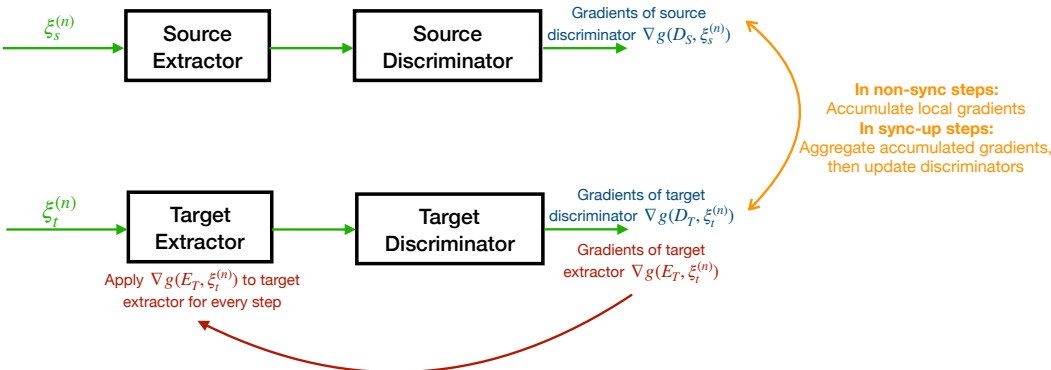

Figure 3: Distributed adversarial training architecture of MDDA

## A.4 Details of Domain Adaptation Algorithms

We now discuss the adversarial training formulation of the various uDA algorithms with which we evaluated the efficacy of our proposed MDDA framework. As shown in Table 4.2, we evaluate our method with four domain adaptation techniques: ADDA (Tzeng et al. (2017)), Gradient Reversal (Ganin et al. (2016)) and Wasserstein DA (Shen et al. (2018)), and CADA (Zou et al. (2019)) Below are the adversarial training formulations of these techniques as proposed in their original papers.

**ADDA**. Following the same notations used earlier in the paper, the adversarial loss formulations of ADDA can be represented mathematically as:

$$\min_D \mathcal{L}_{adv_D} = -\mathbb{E}_{x_s \sim \mathcal{X}_S}[\log(D(E_S(x_s)))] - \mathbb{E}_{x_t \sim \mathcal{X}_T}[\log(1 - D(E_T(x_t)))] \tag{7}$$

$$\min_{E_T} \mathcal{L}_{adv_M} = -\mathbb{E}_{x_t \sim \mathcal{X}_T}[\log(D(E_T(x_t)))] \tag{8}$$

The Discriminator D is optimized using $L_{adv_D}$ where the domain data from source and target domains are assigned different domain labels (0 and 1). To update the feature extractor $E_T$, ADDA proposes to invert the domain labels, which results in the loss formulation given in $L_{adv_M}$.

In order to run ADDA in a distributed manner, we decompose the discriminator $D$ into $D_S$ and $D_T$ which results in the following loss functions:

$$\mathcal{L}_{adv_{D_S}} = -\mathbb{E}_{x_s \sim \mathcal{X}_S}[\log(D_S(E_S(x_s)))] \tag{9}$$

$$\mathcal{L}_{adv_{D_T}} = -\mathbb{E}_{x_t \sim \mathcal{X}_T}[\log(1 - D_T(E_T(x_t)))] \tag{10}$$

$$\min_{E_T} \mathcal{L}_{adv_M} = -\mathbb{E}_{x_t \sim \mathcal{X}_T}[\log(D_T(E_T(x_t)))] \tag{11}$$

Thereafter, we compute local gradients for $D_S$ and $D_T$,

$$\nabla g(D_S, x_s) = \frac{\delta L_{adv_{D_S}}}{\delta D_S} \tag{12}$$

$$\nabla g(D_T, x_t) = \frac{\delta L_{adv_{D_T}}}{\delta D_T} \tag{13}$$

and aggregate them in the sync-up step as shown in Algorithm 1. The aggregated gradients are used to optimize $D_S$ and $D_T$, while $E_T$ is optimized using the loss function in Equation 11.

**Gradient Reversal**. The Gradient Reversal approach uses the same loss formulation for the discriminators $D_S$ and $D_T$ as ADDA as shown in Equations 7, 9, 10.

However, to update the target extractor $E_T$, it leverages the gradient reversal strategy, resulting in the following loss function.

$$\min_{E_T} \mathcal{L}_{adv_M} = -\mathcal{L}_{adv_{DT}}$$
$$= \mathbb{E}_{x_t \sim \mathcal{X}_T}[\log(1 - D_T(E_T(x_t)))] \tag{14}$$

The computation of local gradients for $D_S$ and $D_T$ follow the same process as shown in Equations 12 and 13.

**Wasserstein DA.** In this recently published technique, authors use the Wasserstein distance as the loss function for discriminator. Wasserstein distance between two datasets is defined as

$$\text{Wasserstein}(X_s, X_t) = \frac{1}{n_s} \sum_{x_s \sim \mathcal{X}_S} D(E_S(x_s)) - \frac{1}{n_t} \sum_{x_t \sim \mathcal{X}_T} D(E_T(x_t)) \tag{15}$$

where $n_s$ and $n_t$ are the number of samples in the dataset. The discriminator loss is computed as:

$$\min_{D} \mathcal{L}_{adv_D} = -\mathbb{E}_{x_s \sim \mathcal{X}_S, x_t \sim \mathcal{X}_T}[\text{Wasserstein}(x_s, x_t)] + \gamma L_{grad} \tag{16}$$

where $L_{grad}$ is the gradient penalty used to enforce the Lipschitz constraint on the discriminator. Further, the target extractor is optimized using the following loss function:

$$\min_{E_T} \mathcal{L}_{adv_M} = \mathbb{E}_{x_s \sim \mathcal{X}_S, x_t \sim \mathcal{X}_T}[\text{Wasserstein}(x_s, x_t)] \tag{17}$$

We use the same strategy to compute local gradients for $D_S$ and $D_T$ as shown in Eq 12 and 13.

**CADA.** Consensus Adversarial Domain Adaptation is a technique recently proposed by Zou et al. (2019) which enforces the source and target extractors to arrive at a consensus in the feature space through adversarial training. It uses the same loss formulation for the discriminators $D_S$ and $D_T$ as ADDA as shown in Equations 7, 9, 10. However, the key difference is that CADA optimizes both the source and target feature extractors in the training process, until the discriminator can no longer distinguish the features from source and target domains.

$$\min_{E_S} \mathcal{L}_{adv_{M1}} = -\mathbb{E}_{x_s \sim \mathcal{X}_S}[\log(D_S(E_S(x_s)))] \tag{18}$$

$$\min_{E_T} \mathcal{L}_{adv_{M2}} = -\mathbb{E}_{x_t \sim \mathcal{X}_T}[\log(D_T(E_T(x_t)))] \tag{19}$$

Thereafter, a shared classifier is trained on the sourced labeled data by keeping the source feature extractor fixed. The shared classifier can be used with the target extractor to make predictions.

$$\min_{C_{Shared}} \mathcal{L}_{clf} = -\mathbb{E}_{(x_s,y_s)\sim(X_S,Y_S)} \sum_{k=1}^{K} \mathbb{1}_{[k=y_s]}[\log(C_{Shared}(E_S(x_s))]$$

