# OpenReview forum: "Multi-Step Decentralized Domain Adaptation"
_ICLR.cc/2020/Conference — Reject_

### Official Review · AnonReviewer3 · 2019-10-08
**Official Blind Review #3**

**Rating:** 6

**Review:**

The paper focuses on the problem of domain adaptation among multiple domains when some domains are not available on the same machine. The paper builds a decentralized algorithm based on previous domain adaptation methods.

Pros:
1. The problem is novel and practical. Previous domain adaptation assumes that source and target domains are available but it can happen when the source and target domains have connection issues.
2. The method exploits asynchronizing accumulating and synchronizing update, which reduces the cost of communication between domains.
3. The paper proposes to use Wasserstein distance to select the optimal domain as the source domain for the target domain.
4. The experimental results show that the proposed method outperforms baselines.

Cons:
1. The asynchronizing accumulating and synchronizing update is not novel. It has been used in other communities such as reinforcement learning.

Overall, the paper is good and it is technically sound. The contribution is not significant to the community but providing a new perspective for domain adaptation. I vote for weak accept.

Thank Reviewer1 for reminding. I think the paper still has some novelty and the comments address my concerns. I do  not change my score. Also, I'm not unhappy if the paper is rejected. It is more like a borderline paper.

**Experience Assessment:**

I have published in this field for several years.

**Review Assessment: Checking Correctness Of Derivations And Theory:**

I carefully checked the derivations and theory.

**Review Assessment: Checking Correctness Of Experiments:**

I carefully checked the experiments.

**Review Assessment: Thoroughness In Paper Reading:**

I read the paper thoroughly.

---

> ### Author Response · Authors · 2019-11-13
> **Response to Reviewer 3**
>
> Thanks for finding the problem novel and practical, and for considering our contribution as technically sound.
>
> ### Cons (1) ###
>
> Indeed, as you mentioned, asynchronized accumulating and synchronized updating are variants of existing techniques in distributed training. However, to the best of our knowledge, these techniques have not been applied to adversarial domain adaptation, as such we adopted them to reduce the communication costs associated with sharing discriminator gradients between nodes. We have added this explanation in the paper in the related work section.
>
> Moreover, different from other works and specific to adversarial training, we only accumulate the gradients of parts of the model (source and target discriminators), while the remaining model (target encoder) is updated after every batch without any gradient accumulation (please refer to 3.2 for details). Our experimental evaluation shows a novel finding that this partial accumulation of gradients does not hurt the convergence rate as compared to conventional training of DA algorithms.
>
> Finally, we would like to highlight that the core contribution and novelty of our work lies in the problem formulation and our proposed collaborator selection algorithm. To the best of our knowledge, both these perspectives are missing from current DA literature and therefore, we believe that our work provides a significant contribution to the ML community. These contributions, in combination with our lazy-synchronized distributed training strategy, make our proposed framework useful for real-world deployments of domain adaptation algorithms.

---

### Official Review · AnonReviewer1 · 2019-10-13
**Official Blind Review #1**

**Rating:** 3

**Review:**



###Summary###

This paper tackles unsupervised domain adaptation in a decentralized setting.  The high-level observation is that the conventional unsupervised domain adaptation has two bottlenecks, namely excessive centralization and poor support for distributed domain datasets. The paper proposes Multi-Step Decentralized Domain Adaptation (MDDA) to transfer the knowledge learned from the source domain to the target domain without sharing the data.

The paper also explores explore a proposition: in addition to adapting from the labeled source, can uDA leverage the knowledge from other target domains, which themselves may have undergone domain adaptation in the past.

The proposed MMDA method contains a feature extractor (E), a domain discriminator (D) and task classifier (C) for each domain. The target domain components are initialized with the respective source components. The source domain discriminator D_s target domain discriminator D_t are synchronized by exchanging and averaging the gradients. The paper also proposes Lazy Synchronization to reduce the communication cost of the algorithm.

The paper also proposes Wasserstein distance guided collaborator selection schema to perform the domain adaptation task.

The paper performs experiments on five image and audio datasets: Rotated MNIST, Digits, and Office-Caltech, DomainNet and Mic2Mic.

The baselines used in this paper include "Labeled Source", "Random Collaborator", and "Multi-Collaborator". The experimental results demonstrate that the proposed method can outperform the baselines on some of the experimental settings. The paper also provides a detailed analysis of the model and experimental results.

### Novelty ###

This paper does not propose a new domain adaptation algorithm. However, the paper introduces some interesting tricks to solve the MMDA task such as the lazy synchronization between the source domain discriminator and the target domain discriminator.

###Clarity###

Several critical explanations are missing from the paper:
1) When training the source domain discriminator D_s and target domain discriminator D_t, if the features between the source domain and target domain cannot be shared with each other, how to train the D_s and D_t. For example, the D_s cannot get access to the features from the target domain, how to train D_s?
2) How is the target classifier C_t updated when there are no labels for the target domain?
3) As far as I understand, the domain discriminator is this paper is trained adversarially. The detailed adversarial training step is unclear.

###Pros###

1) The paper proposes an interesting transfer learning schema where the data between the source and target domain can not be shared with each other to protect the data-privacy.

2) The paper provides extensive experiments on multiple standard domain adaptation benchmarks, especially the most recent dataset such as the DomainNet.

3) The paper provides detail empirical analysis to demonstrate the effectiveness of the proposed methods.

###Cons###

1) The most critical issue of this paper is that some explanations are missing, e.g. how are D_s, D_t, C_t trained? Refer to the #Clarity.

2) The presentation and writing of this paper need polish. The author should do more relative surveys to motivate the authors. One critical relevant reference of this paper is:
"Secure Federated Transfer Learning", Yang Liu et al

https://arxiv.org/pdf/1812.03337.pdf

3) The baselines used in this paper is also trivial. It is desirable to compare the proposed method with state-of-the-art domain adaptation methods.

Based on the summary, cons, and pros, the current rating I am giving now is "reject". I would like to discuss the final rating with other reviewers, ACs.
To improve the rating, the author should explain the questions I proposed in the #Clarity

**Experience Assessment:**

I have published in this field for several years.

**Review Assessment: Checking Correctness Of Derivations And Theory:**

I assessed the sensibility of the derivations and theory.

**Review Assessment: Checking Correctness Of Experiments:**

I carefully checked the experiments.

**Review Assessment: Thoroughness In Paper Reading:**

I read the paper thoroughly.

---

> ### Author Response · Authors · 2019-11-13
> **Response to Review 1 (part 1)**
>
>
>
> Thanks for providing a very nice summary of our work and for your insightful comments. We are glad that you found our paper interesting and considered our experimental analysis as extensive and detailed.
>
> ###Novelty###:
>
> We agree with your point that our paper is not proposing a new domain adaptation algorithm to boost the accuracy of the model in the target domain. Instead, our contribution operates one layer above the adaptation algorithm and can be utilized with many existing domain adaptation techniques as we demonstrate in Table 2.
>
> We would like to argue that our proposed contribution is novel – both from a problem formulation and a solution perspective. To the best of our knowledge, no prior domain adaptation paper has looked at the problems of domain selection with distributed domain datasets, which, as we highlight in the paper, are of practical significance but have been overlooked. Our proposed solution consists of the Wasserstein-distance collaborator selection algorithm to find the best possible collaborator for adaptation, and the Lazy-Synchronized DA algorithm to reduce the communication between nodes to merely the gradients of the discriminator (summarized in Algorithm 3).  The former algorithm is particularly noteworthy because the question of ‘which domain should I select to adapt from’ is no less important than ‘how to adapt between two selected domains (which has been mainly studied before)’ in multiple domains adaptation. Overall, we believe our submission provides a brand-new perspective and novel contribution to the domain adaptation literature.
>
> #### Clarity (1) ####
>
> Please correct us if we misunderstood your question -- our interpretation is that you are asking how to train D_s and D_t in a distributed way, when we are not sharing the features extracted from source and target domains across nodes. Indeed, this is a valid question and touches the core of our contribution on distributed training of the discriminators.
>
> To answer this question, let us first have a brief look of how the Discriminator D gets trained in the non-distributed case (Equation 1), where the data from source domain and target domain are fed into their extractors respectively, then the input of D is the concatenation of the output of source extractor (E_s) and target extractor (E_t).
>
> By contrast, as described in Section 3.2 and Algorithm 1, in the distributed case, D is split across nodes as D_s and D_t, and the outputs of E_s and E_t are fed to the respective discriminators separately. As you rightly questioned, it is not possible to update a discriminator without seeing the features from the other domain -- as such, our idea is to
>
> i) compute local gradients for D_s and D_t on their respective nodes (Algo 1, line 4),
> ii) exchange and average the gradients during the sync-up step (Algo 1, line 10),
> iii) update D_s and D_t with the averaged gradients (Algo 1, line 11).
>
> In effect, this guarantees the following:
>
> a) We are able to exchange knowledge between the two domains through sharing gradients of D_s and D_t, without requiring the raw data or extracted features to be shared across nodes.
>
> b) Weights of D_s and D_t always remain identical as they get updated with the same averaged gradients. As such, both the discriminators are always in sync with each other. This means that we are able to keep domain datasets private, and yet perform the adversarial training as given in Equation 1 and 2.
>
> We have added this clarification in the paper in Section 3.2 and also added a figure in the Appendix as a visual aid to explain our distributed training mechanism.
>
> #### Clarity (2) ####
>
> We follow the convention used in past DA papers (e.g. [1]) wherein the target classifier C_t is initialized with C_s (C_t <-- C_s) and is not updated in the training process. The intuition is that if the feature space of the target domain can be successfully aligned with the source domain through adversarial training, i.e., the outputs of E_s and E_t become close enough, then C_s can directly be used in the target domain without any adaptation. Recall that the classifier takes the outputs of the feature extractor as inputs, therefore if feature extractor outputs become similar across the two domains, then the same classifier can be shared by the two domains.
>
> As such, during adversarial training, we only adapt the feature extractor E_t with the goal of aligning the target feature space with the source. In practice, this also means that the classifier C_s or C_t is very simple (e.g., it could be just a softmax layer or one fully-connected layer) and E_t does the bulk of the work for the classification task.
>
> We have added this clarification to the paper in Section 3.2.

---

> > ### Author Response · Authors · 2019-11-13
> > **Response to Reviewer 1 (part 2)**
> >
> > ### Clarity (3) ###
> > Although the adversarial training process is provided in Equation 1 and 2 in Section 3.1, we acknowledge that we did not explain it in detail in the text.
> >
> > Broadly, to train the domain discriminator, we follow the formulation given in equation 1. The key difference is that in our setting, since the source (X_s) and target (X_t) datasets reside in different nodes, we compute the gradients of the discriminators (D_s and D_t) separately and then update them based on their average gradients as discussed in Algorithm 1.
> >
> > Thereafter, discriminators and the target extractor E_t play the adversarial game depending on the underlying domain adaptation algorithm. As shown in Table 2, we evaluated our approach in conjunction with three domain adaptation algorithms:
> >
> > i) ADDA [1] which implements adversarial training by reversing domain labels,
> > ii) GRL [2] which uses a Gradient Reversal Layer between the feature extractor and the discriminator to enable adversarial learning,
> > iii) Wasserstein DA [3] which uses the Wasserstein loss for adversarial training.
> >
> > We have added the details of the adversarial formulation of each algorithm in Section A.3 in the Appendix.
> >
> >
> > ### Cons (2) ###
> >
> > We have polished the paper writing by adding more details and explanations of the design choices and algorithms. We have also revamped the related work section.
> >
> > ### Cons (3) ###
> >
> > Certainly, we will be happy to add results for more DA algorithms in the paper. We however note that due to the novelty of our problem setting and solution, there are no baselines which do collaborator selection, against which we can directly compare our results. Therefore, in this setting, we chose baselines as “Random selecting collaborator (no strategy)”, “Always selecting the first source domain (majority of the prior DA algorithms)” and “Selecting multiple collaborators (a recent work [4])”.  Besides, we have shown the efficacy of our technique with 3 DA [1,2,3] techniques in Table 2.
> >
> > We can extend our analysis (in Table 2) to include an additional (latest) DA algorithm. Would that be sufficient from your perspective?
> >
> > [1] Tzeng et al. Adversarial discriminative domain adaptation. CVPR 2017.
> > [2] Ganin et al. Domain-adversarial training of neural networks.JMLR, 2016.
> > [3] Shen J. et al. Wasserstein distance guided representation learning for domain adaptation. AAAI 2018
> > [4] Zhao et al. Adversarial Multi-Source Domain Adaptation. NeurIPS 2018.

---

### Official Review · AnonReviewer2 · 2019-11-01
**Official Blind Review #2**

**Rating:** 6

**Review:**

This paper focuses on the problem of unsupervised domain adaptation in practical machine learning systems. To address the problems in current unsupervised domain adaptation methods, the authors propose to a novel multi-step framework which works in a decentralized and distributed manner. Experimental results show the effectiveness of the proposed approach.

This paper is well-motivated and the proposed method is novel for unsupervised domain adaptation. The paper is well-supported by theoretical analysis, however, the improvements are not that significant on some experimental results. For the above reasons, I tend to accept this paper but wouldn't mind rejecting it.

Questions:
1. The experiments do not really show the superiority of the proposed method compared to the common centralized approaches as they have similar performances on both collaborator selection and distributed domain adaptation. Can you convince the readers more with some other experiments?
2. What is the weakness of such decentralization models?

**Experience Assessment:**

I have read many papers in this area.

**Review Assessment: Checking Correctness Of Derivations And Theory:**

I assessed the sensibility of the derivations and theory.

**Review Assessment: Checking Correctness Of Experiments:**

I assessed the sensibility of the experiments.

**Review Assessment: Thoroughness In Paper Reading:**

I read the paper at least twice and used my best judgement in assessing the paper.

---

> ### Author Response · Authors · 2019-11-13
> **Response to Reviewer 2**
>
>
>
> Thanks for the review, and for finding our paper well-motivated, novel for unsupervised domain adaptation, and well-supported by theoretical analysis. Here are our responses with respect to your questions:
>
> ### Question 1 ###
>
> We would like to humbly submit that our results indeed demonstrate the superiority of our technique. As you have correctly identified in your review, our paper proposes two techniques: a) *collaborator selection*, i.e., which existing domain should be picked as a collaborator for a new target domain and b) *distributed domain adaptation*, i.e., how to conduct adaptation between two distributed computing nodes without sharing domain data.
>
> For the evaluation of the first technique, we show in Table 1 that in majority of the cases, our collaborator-selection algorithm (MDDA) outperforms conventional strategies used in existing DA works such as always selecting the labeled-source domain as collaborator, selecting multiple collaborators etc.
>
> For example, in the case of RMNIST (O1), MDDA outperforms the top baseline by as much as *33%* in target domain accuracy. There are indeed some cases where our performance is lower than the best performing baseline, but not by much (e.g., 0.84% in Mic2Mic (O2)). On average, it can be easily seen that MDDA significantly outperforms other strategies.  Further, in Table 2, we extend our analysis to demonstrate that our collaborator selection algorithm can work in conjunction with three different DA techniques (ADDA, GRL, Wasserstein DA), and also outperform the existing baselines in all cases.
>
> Once we demonstrated the efficacy of our collaborator selection (our first algorithmic contribution), we proceeded to evaluate the performance of distributed domain adaptation (our second contribution).
>
> It is important to clarify the goal of this evaluation. It is obvious that if we can perform domain adaptation in a distributed manner without requiring the domain datasets to be shared across nodes, it provides certain privacy benefits over non-distributed training. At the same time, there is a tradeoff between the adaptation accuracy and communication costs in distributed DA — if we are willing to incur high communication costs, we can achieve the same accuracy as non-distributed training (by exchanging gradients of each batch). In order to save communication costs, we proposed the Lazy Synchronized DA algorithm which instead exchanges gradients after every p batches.
>
> As such, the primary evaluation goal for our distributed DA algorithm is to reach as close as possible to the accuracy of a conventional non-distributed DA algorithm, with significantly lower communication costs. We have also added this explanation in a concise form in Section 4.
>
> As can be seen in Fig 2 and Table 3, we are indeed able to reach significantly close to the target accuracy obtained with non-distributed DA approaches, even when we synchronize the gradients after every 4 batches. In other words, we incur only 25% of the communication costs of a fully-synchronized algorithm and yet are able to achieve similar levels of target domain accuracy, on top of the privacy benefits that are inherent in a distributed training technique.
>
> Therefore, in summary - we show that our collaborator selection algorithm provides significant accuracy gains over existing baselines. On the other hand, our Lazy-Synchronized training approach -- in addition to enhancing user privacy and saving significant communication costs -- can achieve similar levels of accuracy as conventional non-distributed DA algorithms. We believe that both these results are significant and novel contributions to the DA literature.
>
> ### Question 2 ###
>
> One limitation of our approach is the overhead associated with collaborator selection algorithm. While we showed that selecting an optimal collaborator can significantly boost the target domain accuracy, the selection of such collaborator requires computing the Wasserstein distance for different domain pairs which adds a one-time overhead before the adversarial training begins. Given the benefits associated with collaborator selection (as shown in Table 1 and 2), we believe it is worth paying a small price associated with running the selection algorithm.
>
> Another weakness is that of indirect information leakage in distributed training. In conventional non-distributed DA, users are clearly aware that they are releasing their data over the network, hence giving up on their privacy. However, in distributed training, although there is an impression that the user data is private, prior works have shown the possibility of indirect information leakage [1] and privacy attacks. While a detailed analysis of privacy attacks on our method and their mitigation is beyond the scope of this paper, we have added it as an avenue for future research in the paper.
>
> We hope our responses have answered your questions and convinced you about the significance of our experimental findings.

---

### Official Review · AnonReviewer4 · 2019-11-01
**Official Blind Review #4**

**Rating:** 3

**Review:**

I read the authors response. I am satisfied with the explanations on the privacy party. However, decentralized training part is still unsatisfactory since the empirical evaluation is not really decentralized. Back of the envelope calculations are at best correlated to the actual times spent by each node.  Hence, the numbers in Table 3 are not physical numbers, rather result of an idealized network. Moreover, the x-axis of Figure 2 being training step is still not acceptable. Decentralized and centralized methods should be compared in terms of time which is the only fair metric. I stick to my original decision.

-----

The manuscript is proposing a method for domain adaptation in a private and distributed setting where there are multiple target domains and they are added in a sequential manner. The proposed method considers only the domain adaptation methods in which the source model training and the target model training are done separately. In this setting, existing adapted models can be used as a source domain since a trained model suffices for adaptation. One major contribution of the paper is proposing a straightforward but successful method to choose which domain to adapt from. The main algorithmic tool is estimating Wasserstein distance and choosing the closest domain. The second contribution is distributed training setting for privacy and decentralization.

Choosing which model to adapt from is an interesting contribution. The proposed setting is definitely sensible and the proposed method is theoretically sound. Hence, I consider this as a valuable contribution to the domain adaptation literature. Moreover, results suggest that it also results in significant performance improvement.

Privacy and decentralized learning part has major issues. First of all, the privacy and learning private models is a sub-field of machine learning with a large literature. Authors do not discuss any of these existing work. Second of all, authors do not specify the definition of privacy they are using. Only guarantee the  algorithm provides is not passing data around. However, this is clearly not enough. Passing gradients might result in sharing sensitive data. The actual data can be reconstructed (upto some accuracy) using the gradients passed between nodes. Therefore, either a justification or a privacy guarantee result is needed. Both of these are major issues which need to be fixed.

Decentralized learning is also an important problem which have been studied significantly. Related work section is missing majority of recent and existing work on distributed learning and federated learning. Moreover, empirical study is very counter intuitive. Results are given in terms of accuracy vs number of training steps. The important metrics are amount of massages passed and the total time of the distributed training. Many distributed algorithms trade off having less accurate gradients (consequently having higher number of gradient updates) with less message passing. Hence, I am not sure how to understand the distributed domain adaptation experiments. I am not even sure what the time in Table 3 actually means since it is clearly not even experimented in a distributed setting.

In summary, the submission is addressing an important problem. Moreover, the contribution on collaborator selection is interesting and seems to be working well. However, the private and decentralized learning parts are rather incomplete from related work and experiment sense. Moreover, I am also not sure can we call this method private or not.

**Experience Assessment:**

I have published one or two papers in this area.

**Review Assessment: Checking Correctness Of Derivations And Theory:**

I carefully checked the derivations and theory.

**Review Assessment: Checking Correctness Of Experiments:**

I carefully checked the experiments.

**Review Assessment: Thoroughness In Paper Reading:**

I read the paper at least twice and used my best judgement in assessing the paper.

---

> ### Author Response · Authors · 2019-11-13
> **Related Work and Privacy**
>
>
>
> Thanks for your comments and for finding our problem setting sensible, our approach as theoretically sound and a valuable contribution to the DA literature. Based on your feedback, we have revamped the related work section to include more literature on privacy and decentralized learning.
>
> With regard to privacy in the distributed setting, we highlight that our approach (algorithm 1) only shares the gradients of the **domain discriminators** between nodes. The raw data from both domains as well as its corresponding gradients from the feature extractor are completely private and are NEVER shared between nodes.
>
> This, in turn, reduces the possibility of reconstructing the training data as was demonstrated in earlier private-ML works such as [1], which used the gradients of the classifier to reconstruct the raw data. We will clarify in the paper that this is the level of privacy provided by MDDA in its current form.
>
> To the best of our knowledge, no prior work has shown that discriminator gradients can leak the raw data training data. However, we accept and acknowledge your point and do not discount the possibility that privacy attacks could be developed even on discriminator gradients in the future. While a detailed analysis of privacy and security attacks and their mitigation is out of scope for this paper, we have added a discussion on the possibility of information leakage with our method and pointed to relevant work in the ML privacy literature on the attack defenses.
>
> [1] Aono, Yoshinori, et al. "Privacy-preserving deep learning via additively homomorphic encryption." IEEE Transactions on Information Forensics and Security 13.5 (2017): 1333-1345.

---

> > ### Author Response · Authors · 2019-11-14
> > **On Decentralization and Distributed Training**
> >
> > With regards to the term ‘decentralization’ used in the paper, we would like to clarify that ‘decentralized domain adaptation’ means that target domains can adapt from not only the source domain, but also other target domains, which eliminates the strict dependency on the source domain (which can be seen as a ‘central node’). This is different from the well-known ‘decentralized training’ of machine learning models where the idea is that the information from all distributed workers won’t be aggregated globally by a central node (like parameter server) or message passing process (like All-Reduce). We understand that the different definition of ‘decentralization’ used in the paper might confuse a reader, as such, we have added more clarity about it in the text.
> >
> > To your second question about the metrics used to evaluate our distributed training strategy, we note that our evaluation is done through the lens of domain adaptation as opposed to a general-purpose evaluation of a decentralized training algorithm. In that aspect, our evaluation has two objectives:
> >
> > a) After splitting the discriminator into two replicas (source and target discriminator) and applying lazy-synchronization, can we still achieve similar convergence rate and target domain accuracy as the case where the source and target discriminators reside on the same node (the convention, non-distributed DA case). Figure 2 and Table 3, in fact, confirm that even with a distributed adversarial training mechanism, we can reach similar levels of target domain accuracy as the non-distributed case.
> >
> > b) what is the total time taken for domain adaptation training by our lazy-synchronization approach which exchanges the gradients across nodes for every p batches, when compared to fully-synchronized distributed training (which exchange gradients for each batch) and the non-distributed case (which is conventional in DA literature)? The total training time is the sum of local computation time and communication time across nodes. In the non-distributed case, there is no communication at all during the training, so the training time is just the computation time. But it requires the data from all domains on one node. In the fully synchronized case, the amount of communication equals to the size of discriminator gradients multiplied by the number of training steps, whereas in the lazy-synchronized case, the communication amount is one *p*th of the amount of fully synchronized case, where p is the sync-up step. Based on the amount of data communicated across nodes, we estimate the communication time by assuming a bandwidth of 40Mbps [1] which is roughly the global average upload speed. The computation time is measured for each training step and aggregated to get the total computation time. By adding the total computation time with the total communication time, we obtain the total training time, which we report in Table 3 as one of metrics to evaluate our distributed method.
> >
> > We acknowledge that this description was not adequately provided in the first draft, and based on your feedback, we have now incorporated it in the paper.
> >
> > We would also like to highlight that our approach of estimating training time in a distributed setting is not ideal as it does not take into account the effect of real-world factors such as network setup latency, network congestion etc. However, it does provide a reasonable estimate to compare different domain adaptation training approaches, which can further be verified through a real-world deployment of domain adaptation algorithms on user devices in future work.
> >
> > [1] https://www.speedtest.net/global-index

---

### Author Response · Authors · 2019-11-15
**Summary of changes**

We thank the reviewers for finding our problem setting as sensible, well-motivated, and practical [R2, R3, R4]; our proposed solution as interesting [R1], novel [R2], technically solid [R3], theoretically sound and a valuable contribution to the DA literature [R4], and for considering our experimental analysis as extensive and detailed [R1]. We also appreciate the feedback and constructive critique from all the reviewers, which has helped us in significantly improving our paper.

Below is a summary of the major changes we have made in the paper:

***Based on the feedback from R1***

- We have substantially improved the description of our distributed training mechanism in Section 3.2.
- We have described the adversarial learning formulations used in our experiments in the Appendix.
- We conducted a larger literature survey and extended the related work section.
- We have added a new domain adaptation technique in Table 2 to demonstrate that our approach can work with the latest DA techniques. This specific baseline called CADA is from a work published at AAAI 2019 and the authors show that it outperforms many recent uDA techniques.

***Based on the feedback from R2***

- In Section 4, we have improved the writing to ensure that the objective of various evaluations done in the paper is clear.
- We have added the limitations of our techniques while describing them, as well as summarized them in a separate section (Limitations and Conclusion).

***Based on the feedback from R3***

- We have improved the description of our distributed training algorithm in Section 3.2 and also added a figure of our training architecture in the Appendix.
- We have added prior works on decentralized and distributed training in the Related Work and highlighted that our contribution lies in enabling adversarial domain adaptation to work in a distributed manner.

***Based on the feedback from R4***

- We added a detailed discussion about the scope of privacy guarantees of our approach in Section 3.
- We have substantially enhanced the related work section by adding literature on decentralized training and privacy in ML.
- We have clarified the goals of our evaluation in Section 4.2 and clearly explained the metrics used to evaluate our approach. - We have also updated the training time in Table 3 and explained it in the text.
- We have acknowledged the limitations of our approach while introducing them, as well as briefly in Section 5.

---

### Decision · Program_Chairs · 2019-12-19

**Decision:**

Reject

**Comment:**

This paper proposes a solution to the decentralized privacy preserving domain adaptation problem. In other words, how to adapt to a target domain without explicit data access to other existing domains. In this scenario the authors propose MDDA which consists of both a collaborator selection algorithm based on minimal Wasserstein distance as well as a technique for adapting through sharing discriminator gradients across domains.

The reviewers has split scores for this work with two recommending weak accept and two recommending weak reject. However, both reviewers who recommended weak accept explicitly mentioned that their recommendation was borderline (an option not available for ICLR 2020). The main issues raised by the reviewers was lack of algorithmic novelty and lack of comparison to prior privacy preserving work. The authors agreed that their goal was not to introduce a new domain adaptation algorithm, but rather to propose a generic solution to extend existing algorithms to the case of privacy preserving and decentralized DA.  The authors also provided extensive revisions in response to the reviewers comments. Though the reviewers were convinced on some points (like privacy preserving arguments), there still remained key outstanding issues that were significant enough to cause the reviewers not to update their recommendations.

Therefore, this paper is not recommended for acceptance in its current form. We encourage the authors to build off the revisions completed during the rebuttal phase and any outstanding comments from the reviewers.